# Risk of coronary heart disease in the rural population in Xinjiang: A nested case-control study in China

**Changjing Li**[1⊗], **Rulin Ma**[1⊗], **Xianghui Zhang**[1], **Jiaolong Ma**[1], **Xinping Wang**[1], **Jia He**[1], **Jingyu Zhang**[1], **Kui Wang**[1], **Yunhua Hu**[1], **Hongrui Pang**[1], **Lati Mu**[1], **Yizhong Yan**[1], **Yanpeng Song**[1], **Heng Guo**[1]*, **Shuxia Guo**[1,2]*

**1** Department of Public Health, Shihezi University School of Medicine, Shihezi, China, **2** Department of Pathology and Key Laboratory of Xinjiang Endemic and Ethnic Diseases (Ministry of Education), Shihezi University School of Medicine, Shihezi, China

⊗ These authors contributed equally to this work.
* gsxshzu@sina.com (SG); guoheng@shzu.edu.cn (HG)

## Abstract

### Background and aim

Coronary heart disease (CHD) is a chronic complex disease caused by a combination of factors such as lifestyle behaviors and environmental and genetic factors. We conducted this study to evaluate the risk factors affecting the development of CHD in Xinjiang, and to obtain valuable information for formulating appropriate local public health policies.

### Method

We conducted a nested case-control study with 277 confirmed CHD cases and 554 matched controls. The association of the risk factors with the risk of CHD was assessed using the multivariate Cox proportional hazard model. Multiplicative interactions were evaluated by entering interaction terms in the Cox proportional hazard model. The additive interactions among the risk factors were assessed by the index of additive interaction.

### Results

The risk of CHD increased with frequent high-fat food consumption, dyslipidemia, obesity, and family history of CHD after adjustment for drinking, smoking status, hypertension, diabetes, family history of hypertension, and family history of diabetes. We noted consistent interactions between family history of CHD and frequent high-fat food consumption, family history of CHD and obesity, frequent high-fat food consumption and obesity, frequent high-fat food consumption and dyslipidemia, and obesity and dyslipidemia. The risk of CHD events increased with the presence of the aforementioned interactions.

### Conclusions

Frequent high-fat food consumption, family history of CHD, dyslipidemia and obesity were independent risk factors for CHD, and their interactions are important for public health interventions in patients with CHD in Xinjiang.

**Data Availability Statement:** All relevant data are within the manuscript and its Supporting Information files.

**Funding:** This study is supported by National Natural Science Foundation of China (No. 81560551).Shuxia Guo received this award.

**Competing interests:** The authors have declared that no competing interests exist.

## Introduction

Globally, cardiovascular disease is the most common cause of death. CHD accounted for more than 53% of all CVD-related deaths in 2016, and CHD-related mortality rate increased by 19.0% from 2006 to 2016[1]. As the prevalence of risk factors has been increasing, the mortality rate of CHD has increased by 40.1% during the same period, with CHD accounting for 17.8% of all deaths in China [2]. CHD development has become a public health problem that must be addressed urgently. The pathology underlying CHD is coronary arteriosclerosis (AS), which affects the blood circulation and causes myocardial ischemia and hypoxia, leading to a series of vascular diseases. Recently, several studies reported that inflammation, lipid metabolism disorders, and oxidative stress may promote the development of AS[3–5]. However, AS is caused by a combination of factors. Hence, the identification of risk factors and biomarkers plays a vital role in the prevention and treatment of CHD.

Xinjiang is a multi-ethnic province in the northwestern region of China. In Xinjiang, the ethnic minority with the largest number of members is the Uighurs, followed by the Kazakhs. As these groups are economically backward, and the regional healthcare resources are unevenly distributed, there is a lack of cohort studies analyzing the local public health issues, such as the incidence of CHD. The unique ethnic dietary habits and the living environment of these groups are different from those of ethnic groups in China (for example, consumption of a high-salt high-fat high-carbohydrates diet and making their living as nomads). A previous study reported that the prevalence of overweight and obesity is higher among the population in Xinjiang than in the national population in China [6]. Thus, we conducted this nested case-control study by analyzing the biochemical indicators and physical examination results to identify the risk factors of CHD for the population in Xinjiang. To identify potential interactions among these risk factors and to adopt timely and effective preventive measures for the population in Xinjiang.

## Materials and methods

### Study design

The survey among the Kazakhs and the Uyghurs began in April 2009 and was followed up in December 2013, April 2016, and August 2017 in the Xinyuan County and Jiashi County. We selected two representative areas (i.e., Yili and Kashi) according to the distribution of the populations in the Xinjiang Uygur Autonomous Region and randomly selected a county (i.e., Xinyuan and Jiashi Counties) and a township in each county (i.e., Nalati and Jiangbazi Townships). Next, the corresponding villages (six villages in the Nalati Township and 12 villages in the Jiangbazi Township) were selected using the stratified sampling method. Finally, we conducted interviews with 7,258 participants aged > 18 years who lived in the villages for at least 6 months (3,542 Uighurs and 3,716 Kazakhs). All participants provided informed consent. The total response rate was 87.5% (88.6% among the Uygur and 86.5% among the Kazakh). The survey was approved by the Ethical Review Board of the First Affiliated Hospital of Shihezi University School of Medicine (IERB No. SHZ2010LL01).

### Selection of cases and controls

Cases were selected from among the participants who experienced their first CHD event during the follow–up. The presence of CHD was determined on the basis of self–reported questionnaire responses, medical insurance records, and local hospital discharge records from 2009 to 2017. Patients with self-reported CHD findings were required to have a certificate of CHD diagnosis from the medical institution of their township during investigation. Multiple

CHD events may occur in the same patient; the first occurrence was considered as the outcome event. Patients diagnosed with a CHD event were required meet any of the following criteria: hospitalized for CHD; coronary artery atherosclerosis; coronary interventional therapy; angina pectoris; myocardial infarction; and sudden cardiac death. The controls were selected from among the participants without CHD during the survey period, who provided blood samples and information on their physical characteristics. For each case, we selected two controls, which were matched by sex, age, and ethnicity used propensity score matching [7]. Once the controls were matched for a particular case, they could not be selected as controls for other cases.

## Epidemiological survey and biochemical measurements

Data of participants were collected during the family interview using a self-made question-naire. The detailed information on the questionnaire included demographics, diets, drinking and smoking status and personal and family history of the disease. The data obtained from the physical examination included weight, height, and waist circumference (WC), which were measured by trained investigators using a uniform standardized method during the interview. The abovementioned measurement and blood sample collection methods have been previ-ously described [8]. The biochemical parameters from the blood samples included the total cholesterol (TC), triglyceride (TG), low-density lipoprotein cholesterol (LDL-C), high–density lipoprotein cholesterol (HDL-C). All blood samples were analyzed using an automatic bio-chemical analyzer (Olympus AU 2700; Olympus Diagnostics, Hamburg, Germany).

## Definition

Patients whose parents or siblings experienced the CHD were considered to a family history of the CHD. Dyslipidemia was defined as the presence of abnormal levels of TC, TG, low HDL-C and/or high LDL-C. The abnormal TC level was defined as TC $\geq$ 6.2 mmol/l; abnormal TG level, TG $\geq$ 2.3 mmol/l; high LDL-C level, LDL-C $\geq$ 4.1 mmol/l; and low HDL-C level, HDL-C < 1.0 mmol/l[9]. The abnormal WC level in Asians was defined as WC $\geq$ 90 cm for males and WC $\geq$ 80cm for females [10]. The body mass index (BMI) was calculated using the following equation: [weight (kg)] / [height (m) $^\wedge$ 2]. Overweight was defined as BMI $\geq$ 24.0 kg/m$^2$ and obesity is BMI $\geq$ 28.0 kg/m$^2$ [11].

Local high-fat foods include the following: Zhuafan is stewed with fresh lamb, carrots, rice, onions, mutton fat and a large amount of oil (approximately 371 kcal per 100 gm); Baoershake is a fried flour–based food item (approximately 399 kcal per 100 g,); and ghee, a local dairy product similar to butter, is extracted from milk and contains more than 90% saturated fat (approximately 860 kcal per 100 gm), and is always consumed with naan and milk tea. We defined frequent high-fat food consumption as conformance with any of the following criteria: (1) eating 4 servings or more of Zhuafan per week (1 serving is equivalent to 1 bowl of 8 cm caliber or 1 plate of 16 cm caliber), (2) eating Baoershake 4 times or more per week, or (3) eat-ing 5 scoops or more of ghee per day.

## Statistical analysis

Significant differences in the baseline characteristics between the cases (newly developed CHD during follow–up) and controls were compared using the $\chi^2$ test for categorical variables. A multivariate Cox proportional hazard model was used to evaluate the association between the underlying influencing factors and risk of CHD by calculating the hazard ratio (HR) and the associated 95% confidence interval (CI). The survival time was defined as the period from the time the participants were enrolled in the cohort study to the time the first CHD event

occurred. To match the cases and controls (1:2), propensity score matching was performed using Empower Stats statistical software (http://www.empowerstats.com). Data were analyzed using Statistical Product and Service Solutions (SPSS) version 24.0 (Chicago, Illinois, USA). All statistical tests were two–sided, and $P < 0.05$ was considered statistically significant. Multiplicative synergy index (SI) was used to evaluate multiplicative interaction, which is calculated by creating and entering the multiplicative interaction terms in the Cox proportional hazard model. There was no multiplicative interaction if the SI equal 1. Additive interactions were evaluated using three indices, namely, relative excess risk due to interaction (RERI), attributable proportion of interaction (AP), and additive SI, which were calculated using Andersson's Excel calculation table. There was no additive interaction if the following conditions occurred: the 95% CI of the RERI or AP equal 0 and that of the SI equal 1.

## Results

### Baseline characteristics of the study participants

The initial cohort included 7,258 individuals. We further excluded 143 participants with incomplete basic, blood sample, physical information, or patients with a history of CHD. Among the 7115 participants (44,431.65 person years), 277 had their first CHD event during the follow-up period. The incidence of the first CHD event was 55.73/10,000 person-years.

Table 1 summarizes the results of the comparison of the baseline characteristics between the controls (n = 554) and cases (n = 277); the cases included a significantly higher proportion

**Table 1. Baseline characteristics of study participants in the nested case-control study.**

| Characteristics | | Control (n = 554) | Case (n = 277) | $\chi^2$ | $P$ |
|---|---|---|---|---|---|
| WC, n (%) | Normal | 234 (42.24) | 84 (30.32) | | |
| | Abnormal | 320 (57.76) | 193 (69.68) | 11.095 | 0.001* |
| TC, n (%) | Normal | 536 (96.75) | 256 (92.42) | | |
| | Abnormal | 18 (3.25) | 21 (7.58) | 7.748 | 0.005* |
| TG, n (%) | Normal | 485 (85.55) | 235 (84.84) | | |
| | Abnormal | 69 (12.45) | 42 (15.16) | 1.170 | 0.279 |
| LDL-C, n (%) | Normal | 539 (97.29) | 265 (95.67) | | |
| | Abnormal | 15 (2.71) | 12 (4.33) | 1.550 | 0.213 |
| HDL-C, n (%) | Normal | 421 (75.99) | 195 (70.40) | | |
| | Abnormal | 133 (24.01) | 82 (29.60) | 3.015 | 0.083 |
| High-fat food, n (%) | No | 399 (72.02) | 173 (62.45) | | |
| | Yes | 155 (27.98) | 104 (37.55) | 7.878 | 0.005* |
| Family history of CHD, n (%) | No | 533 (96.21) | 249 (89.89) | | |
| | Yes | 21 (3.79) | 28 (10.11) | 13.283 | <0.001** |
| Dyslipidemia, n (%) | No | 324 (58.48) | 123 (44.40) | | |
| | Yes | 230 (41.52) | 154 (55.60) | 14.727 | <0.001** |
| Obesity, n (%) | No | 464 (83.75) | 198 (71.48) | | |
| | Yes | 90 (16.25) | 79 (28.52) | 17.173 | <0.001** |
| Overweight, n (%) | No | 405 (73.10) | 199 (71.84) | | |
| | Yes | 149 (26.90) | 78 (28.16) | 0.148 | 0.700 |

WC, waist circumference; TC total cholesterol; High-fat food, frequent high-fat food consumption; CHD, Coronary heart disease.

* $P < 0.05$,

** $P < 0.001$ for $\chi^2$ test.

of patients who had abnormal WC and TC levels, family history of CHD, dyslipidemia, and obesity and frequent high-fat food consumption (*P* <0.05). There were no significant differences in the high TG, high LDL-C, and low HDL-C levels and the prevalence of overweight.

## Association between the risk factors and CHD

The risk of CHD increased significantly and was associated with abnormal WC levels (HR 1.63, 95%CI 1.26–2.11) and abnormal TC levels (HR 1.59, 95%CI 1.02–2.48), frequent high-fat food consumption (HR 1.83, 95%CI 1.44–2.34), dyslipidemia (HR 1.81, 95%CI 1.43–2.29), obesity (HR 2.02, 95%CI 1.56–2.63), and family history of CHD (HR 2.10, 95%CI 1.42–3.11) in the univariate model (*P* <0.05; **Table 2**). The adjusted HRs in the multivariate model were similar to those in the univariate model for frequent high-fat food consumption, dyslipidemia, obesity, and family history of CHD. We did not observe association of CHD risk with WC and TC levels.

## Multiplicative interaction between the risk factors and CHD

The coexistence of factors increased the risk of CHD compared to that in the reference group, as follows: frequent high-fat food consumption and obesity (HR 3.14, 95%CI 2.19–4.50), frequent high-fat food consumption and family history of CHD (HR 3.64, 95%CI 2.26–5.87), frequent high-fat food consumption and dyslipidemia (HR 2.99, 95%CI 2.18–4.10), obesity and family history of CHD (HR 4.62, 95%CI 2.57–8.31), obesity and dyslipidemia (HR 3.12, 95%CI 2.25–4.32), and dyslipidemia and family history of CHD (HR 2.81, 95%CI 1.66–4.76, *P* <0.001). According to multiplicative SI, the effect of multiplicative interactions between any two factors on CHD had no statistical significance (**Table 3**).

## Additive interaction between the risk factors and CHD

The evaluation indices of the additive interactions are shown in **Table 4**. Among the participants with concurrent exposure to frequent high-fat food consumption and family history of

**Table 2. The results of Cox proportional hazard model for risk factors and CHD.**

| Risk factors | HR [a] (95%CI) | HR [b] (95%CI) | HR [c] (95%CI) |
|---|---|---|---|
| WC | 1.63 (1.26–2.11) [**] | 1.61 (1.24–2.09) [**] | 1.24 (0.93–1.65) |
| TC | 1.59 (1.02–2.48) [*] | 1.54 (0.98–2.41) | 1.22 (0.77–1.94) |
| High-fat food | 1.83 (1.44–2.34) [**] | 1.78 (1.38–2.28) [**] | 1.53 (1.18–1.98) [**] |
| Dyslipidemia | 1.81(1.43–2.29) [**] | 1.80 (1.41–2.29) [**] | 1.51 (1.17–1.96) [**] |
| Obesity | 2.02 (1.56–2.63) [**] | 1.97 (1.51–2.59) [**] | 1.52 (1.13–2.05) [*] |
| Family history of CHD | 2.10 (1.42–3.11) [**] | 2.02 (1.35–3.02) [**] | 1.65 (1.10–2.50) [*] |

WC, abnormal waist circumference levels; TC, abnormal total cholesterol levels; High-fat food, frequent high-fat food consumption; CHD, Coronary heart disease. HR, hazard ratio; CI, confidence interval.

[a] The hazard ratio of Cox proportional hazard model with no adjustment;

[b] The hazard ratio of Cox proportional hazard model with adjustment for drinking, smoking, hypertension, diabetes, family history of hypertension, and family history diabetes;

[c] The hazard ratio of multi-factor Cox proportional hazard model with adjustment for drinking, smoking, hypertension, diabetes, the family history of hypertension, diabetes, and the other risk factors (i.e. abnormal WC levels, abnormal TC levels, frequent high-fat food consumption, dyslipidemia, obesity, family history of CHD);

[*] *P* < 0.05,

[**] *P* < 0.001 for Cox proportional hazard model.

**Table 3. The results of multiplicative interaction for risk factors and CHD.**

| Multiplicative interaction items | HR [b] (95%CI) | Multiplicative SI [b] (95%CI) |
|---|---|---|
| High-fat food (−) / Obesity (−) | 1 (ref) | 1.18 (0.69–2.01) |
| High-fat food (+) / Obesity (−) | 1.57 (1.16–2.11) ** | |
| High-fat food (−) / Obesity (+) | 1.70 (1.17–2.47) ** | |
| High-fat food (+) / Obesity (+) | 3.14 (2.19–4.50) ** | |
| High-fat food (−) / Family history of CHD (−) | 1 (ref) | 1.95 (0.79–4.87) |
| High-fat food (+) / Family history of CHD (−) | 1.61 (1.23–2.10) ** | |
| High-fat food (−) / Family history of CHD (+) | 1.16 (0.54–2.51) | |
| High-fat food (+) / Family history of CHD (+) | 3.64 (2.26–5.87) ** | |
| High-fat food (−) / Dyslipidemia (−) | 1 (ref) | 1.69 (1.00–2.84) |
| High-fat food (+) / Dyslipidemia (−) | 1.23 (0.82–1.86) | |
| High-fat food (−) / Dyslipidemia (+) | 1.43 (1.06–1.94) * | |
| High-fat food (+) / Dyslipidemia (+) | 2.99 (2.18–4.10) ** | |
| Obesity (−) / Family history of CHD (−) | 1 (ref) | 1.51 (0.67–3.40) |
| Obesity (+) / Family history of CHD (−) | 1.84 (1.38–2.47) ** | |
| Obesity (−) / Family history of CHD (+) | 1.66 (0.97–2.84) | |
| Obesity (+) / Family history of CHD (+) | 4.62 (2.57–8.31) ** | |
| Obesity (−) / Dyslipidemia (−) | 1 (ref) | 1.84 (0.99–3.42) |
| Obesity (+) / Dyslipidemia (−) | 1.17 (0.69–1.99) | |
| Obesity (−) / Dyslipidemia (+) | 1.45 (1.09–1.92) * | |
| Obesity (+) / Dyslipidemia (+) | 3.12 (2.25–4.32) ** | |
| Dyslipidemia (−) / Family history of CHD (−) | 1 (ref) | 0.58 (0.26–1.29) |
| Dyslipidemia (+) / Family history of CHD (−) | 1.84 (1.43–2.37) ** | |
| Dyslipidemia (−) / Family history of CHD (+) | 2.65 (1.41–4.95) * | |
| Dyslipidemia (+) / Family history of CHD (+) | 2.81 (1.66–4.76) ** | |

High-fat food, frequent high-fat food consumption; CHD, Coronary heart disease. HR, hazard ratio; CI, confidence interval; SI, synergy index.

[b] Adjustment for drinking, smoking, hypertension, diabetes, family history of hypertension, and family history diabetes;

* $P < 0.05$,

** $P < 0.001$ for multiplicative interaction.

**Table 4. The results of additive interaction for risk factors and CHD.**

| Interactive items | Additive interaction | | |
|---|---|---|---|
| | RERI [b] (95%CI) | AP [b] (95%CI) | Additive SI [b] (95%CI) |
| Family history of CHD & High-fat food | 7.01 (-0.76–14.78) | 0.84 (0.65–1.03) * | 22.70 (0.67–766.70) |
| Family history of CHD & Obesity | 8.12 (-6.02–22.26) | 0.74 (0.37–1.10) * | 5.22 (1.04–26.12) * |
| Family history of CHD & Dyslipidemia | -1.70 (-6.66–3.25) | -0.48 (−2.08–1.18) | 0.60 (0.15–2.45) |
| High-fat food & Obesity | 6.43 (-0.96–13.82) | 0.82 (0.60–1.04) * | 15.42 (0.96–248.68) |
| High-fat food & Dyslipidemia | 1.16 (0.15–2.17) * | 0.46 (0.16–0.76) * | 4.03 (0.59–27.65) |
| Obesity & Dyslipidemia | 1.64 (0.22–3.07) * | 0.52 (0.21–0.83) * | 4.03 (0.75–21.55) |

High-fat food, frequent high-fat food consumption; CHD, Coronary heart disease; RERI, relative excess risk due to interaction; AP, the attributable proportion of interaction; SI, synergy index.

[b] Adjustment for drinking, smoking, hypertension, diabetes and family history of hypertension, diabetes;

* $P < 0.05$ for additive interaction.

CHD, approximately 84% of the risk of CHD was attributed to their interaction effect (AP 0.84, 95%CI 0.65–1.03), and the attributable proportion of interaction was 82% between frequent high-fat food consumption (AP 0.82, 95%CI 0.60–1.04). On the basis of the indices of the additive interactions between frequent high-fat food consumption and dyslipidemia, the relative excess risk was 1.16 (RERI 1.16, 95%CI 0.15–2.17), and the AP was 46% (AP 0.46, 95% CI 0.16–0.76). The RERI between obesity and dyslipidemia was 1.64 (RERI 1.64, 95%CI 0.22–3.07), and AP was 52% (AP 0.52, 95% CI 0.21–0.83). The additive SI among the patients with both family history of CHD and obesity who developed CHD was 5.22 (additive SI 5.22, 95% CI 1.04–26.12), and attributable proportion of interaction was 74% (AP 0.74, 95% CI 0.37–1.10). The statistically significant additive interaction indicators were not noted between family history of CHD and dyslipidemia.

## Discussion

In this nested case-control study based on the population in Xinjiang, we found that frequent high-fat food consumption, family history of CHD, dyslipidemia, and obesity were the independent risk factors of CHD in the multivariate Cox proportional hazard model. Due to the complex relationship between heredity and environmental factors, AS can aggravate oxidative stress, inflammation and metabolic abnormalities involving cholesterol and lipoproteins, thus, leading to CHD. A positive association between obesity and CHD was found in a rural Chinese population. This may be because accumulation of visceral fat promotes insulin resistance[12]. Cohort studies in the Asia-pacific region have reported that dyslipidemia is a definitive cause of arteriosclerosis[13].

A prospective study on the effects of a 6-month low-salt low-fat diet and aerobic exercise found that many risk factors improved and the risk of CHD in women decreased from 6% to 4% and from 16% to 13% in men [14]. Another study on men reported that high-fat and hazelnut-enriched diets are superior to the low-fat control diet, which may have beneficial effects on the risk of CHD owing to the favorable changes in the plasma lipid mass spectrum in adult men[15]. There is probable evidence regarding the importance of diets in the development of CHD, and suitable dietary paradigms must consider the type and quality of fat and carbohydrates required for metabolism by changing the intermediate risk factors ameliorating the characteristics of individual susceptibility [16]. In this study, frequent high-fat food consumption increased the risk of CHD (HR 1.53, 95%CI 1.18–1.98). A previous study reported that high unsaturated fatty acid intake may promote oxidative stress and increase the risk of CHD [17]; further, increased intake of high-fat diet increases the risk of myocardial infarction [18]. However, the relationship remains controversial[19]; a positive association was found between frequent high-fat food consumption and CHD risk in the USA, but not in Europe or Asia [20]. This could be attributed to the different dietary patterns and social conditions, as well as ethnic differences.

Familial aggregation of CHD has been demonstrated, and a family history of CHD is considered a risk factor even after adjusting for other risk factors in a previous study, which is consistent with our data[21]. Notably, CHD is affected by several of environmental and genetic factors; however, family members do not encounter identically similar environmental factors, such as dietary pattern and lifestyle. Therefore, it is necessary to conduct studies on the interaction between genetic and environmental factors.

Several epidemiologists contend that the assessment of biological interactions should involve analysis of additive interactions rather than multiplicative interactions[22]. We analyzed the multiplicative and additive interactions among the risk factors, and some significant additive interaction index were observed. We found a positive additive effect between

dyslipidemia and obesity (RERI 1.64, 95%CI 0.22–3.07; AP 0.52, 95%CI 0.21–0.83). A recent cohort study reported a positive interaction between mixed dyslipidemia and obesity increased the risk of atherogenic[23].

A study reported that the interaction between family history of premature CHD and ≥ 2 metabolic risk factors (e.g., age, male sex, hypertension, dyslipidemia, low HDL-C and high TG levels, smoking status [current], and obesity) amplifies the risk for coronary arterial calcium deposition [24]. This study indicated that interaction between family history of CHD and obesity increases the risk of CHD (AP 0.74, 95%CI 0.37–1.10; additive SI 5.22, 95%CI 1.04–26.12). Based on the results of this study, an interaction between family history of CHD and dyslipidemia was not observed. However, a Swedish study supports the interactions between family history of CHD and LDL/HDL quotient ≥ 4.0 in women for myocardial infarction (SI 3.8, 95% CI 1.5–9.7), this indicates that further research is needed on different indicators of dyslipidemia [25]. Slattery et al. [26] found that patients with family history of CHD show higher levels of fatty foods consumption than those without family history of CHD among a population of older women. We found an interaction between family history of CHD and frequent high-fat food consumption (AP 0.84, 95%CI 0.65–1.03); however, the available evidence is insufficient to demonstrate the existence of these interactions and further research is warranted. The available evidence suggests that family history of CHD is undeniably an independent risk factor for CHD and important factor to help prevent poor CHD outcomes and promote cardiovascular health [27], which is an important factor in the study of CHD in the Xinjiang population.

Although the interactions between frequent high-fat food consumption and obesity, and the interactions between frequent high-fat food consumption and dyslipidemia have rarely been reported, Chmurzynska et al reported that the frequency of consumption of high-fat foods was correlated in the obesity [28]. Moreover, studies have shown that the interaction between a low-fat diet, such as a Mediterranean diet, and genetic factors have impact on the risk for CHD [29, 30]. Further research and evaluation of the interactions between dietary and genetic factors and between dietary and environmental factors are required.

Studying the risk factors of and interactions in CHD plays important roles in reducing the incidence of CHD and improving the quality of life of the patients. Among this cohort, many individuals experience dyslipidemia and obesity, which are known risk factors for CHD, and frequent high-fat food consumption deserves attention as an important factor of CHD. These risk factors can be improved via lifestyle changes. This could help provide important references for developing effective and feasible interventions for patients with CHD in Xinjiang. However, the prevention strategy of combining family history of CHD with obesity, dyslipidemia, and frequent high-fat food consumption must be investigated further.

The present study has some limitations. First, only serological and physical indicators were assessed; these indicators are not sufficient to assess the risk factors of CHD comprehensively, and we did not appropriately evaluate all the effects of interactions between factors on CHD. Second, although we performed face-to-face interview questionnaires-based surveys, bias from self-reported methods is possible. As the majority of male Kazakhs are nomads and were not admitted to hospitals, all relevant disease information may not have been collected. And the overall response rate in our study was 87.5%, which means that the baseline population have not been entirely followed up, and we cannot confirm the disease status of the baseline population who were not followed up. This means that there may be selection bias. Therefore, the present analysis may have underestimated the cumulative incidence of CHD. Third, we did not measure the ratio of fat in the daily total energy and daily dietary fat intake to evaluate the high-fat food consumption; we only used the frequency of high-fat food consumption as clues. Fourth, history of malignancy was not excluded. Despite the above limitations, the study also

has some strengths. First, this study was a long-term cohort study on CHD conducted in the Kazakh and Uygur populations in Xinjiang. A nested case-control study design was adopted in this study to control for confounding factors, such as sex, age, and ethnicity; this would further improve the test efficiency. Second, the routine physical examination results were collected from national health check projects supported by the government, which are comprehensive and highly reliable. Next, only a few studies have reported multiplicative and additive interactions. The abovementioned additive scale is more suitable for assessing the biological interactions.

## Supporting information

**S1 Fig. Histogram of minority between case and control by SPSS.**
(DOCX)

**S2 Fig. Histogram of sex between case and control by SPSS.**
(DOCX)

**S3 Fig. Histogram of age between case and control by SPSS.**
(DOCX)

**S1 Table. Description of matching factors between case and control by SPSS (minority).**
(DOCX)

**S2 Table. Description of matching factors between case and control by SPSS (sex).**
(DOCX)

**S3 Table. Description of matching factors between case and control by SPSS (age).**
(DOCX)

**S1 Questionnaire.**
(DOCX)

**S1 Data.**
(SAV)

## Acknowledgments

We sincerely thank all participated who in the study and the clinical laboratory of First Affiliated Hospital of Shihezi University School of Medicine for their work.

## Author Contributions

**Conceptualization:** Changjing Li, Rulin Ma.

**Data curation:** Changjing Li, Jia He.

**Formal analysis:** Changjing Li.

**Funding acquisition:** Shuxia Guo.

**Investigation:** Changjing Li, Rulin Ma, Xianghui Zhang, Jiaolong Ma, Xinping Wang, Jia He, Jingyu Zhang, Yunhua Hu, Hongrui Pang, Lati Mu, Yanpeng Song.

**Methodology:** Changjing Li, Kui Wang.

**Project administration:** Shuxia Guo.

**Resources:** Shuxia Guo.

**Software:** Changjing Li, Xianghui Zhang, Jingyu Zhang.

**Supervision:** Changjing Li, Shuxia Guo.

**Validation:** Jiaolong Ma, Xinping Wang, Yizhong Yan.

**Visualization:** Xinping Wang, Yunhua Hu.

**Writing – original draft:** Changjing Li, Rulin Ma, Heng Guo, Shuxia Guo.

**Writing – review & editing:** Changjing Li, Rulin Ma, Heng Guo, Shuxia Guo.

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
