## [Decision Letter · Decision Letter 0]

17 Dec 2019

PONE-D-19-21029

Risk of coronary heart disease among Xinjiang rural: a nested case-control study in China

PLOS ONE

Dear Dr. Guo

Thank you for submitting your manuscript to PLOS ONE. After careful consideration, we feel that it has merit but does not fully meet PLOS ONE’s publication criteria as it currently stands. Therefore, we invite you to submit a revised version of the manuscript that addresses the points raised during the review process.

We would appreciate receiving your revised manuscript by February 1, 2020. To enhance the reproducibility of your results, we recommend that if applicable you deposit your laboratory protocols in protocols.io, where a protocol can be assigned its own identifier (DOI) such that it can be cited independently in the future. For instructions see: http://journals.plos.org/plosone/s/submission-guidelines#loc-laboratory-protocols

We look forward to receiving your revised manuscript.

Kind regards,

Alana T Brennan

Academic Editor

PLOS ONE

Journal Requirements:

1. Please include additional information regarding the survey or questionnaire used in the study and ensure that you have provided sufficient details that others could replicate the analyses. If you developed a questionnaire as part of this study and it is not under a copyright more restrictive than CC-BY, please include a copy, in both the original language and English, as Supporting Information.

2.

We suggest you thoroughly copyedit your manuscript for language usage, spelling, and grammar. If you do not know anyone who can help you do this, you may wish to consider employing a professional scientific editing service.  

Reviewers' comments:

Reviewer's Responses to Questions

**Comments to the Author**

1. Is the manuscript technically sound, and do the data support the conclusions?

Reviewer #1: Partly

Reviewer #2: Partly

2. Has the statistical analysis been performed appropriately and rigorously? 

Reviewer #1: N/A

Reviewer #2: I Don't Know

3. Have the authors made all data underlying the findings in their manuscript fully available?

Reviewer #1: No

Reviewer #2: Yes

4. Is the manuscript presented in an intelligible fashion and written in standard English?

Reviewer #1: No

Reviewer #2: No

5. Review Comments to the Author

Reviewer #1: Dear manuscript authors,

This paper presents the potential interactions among the main risk factors of CHD. It helps to clarify the risk factors of CHD for the population in Xinjiang.

This study presents some original research and they are important only for this region. Some definitions, for example, obesity, overweight are different compared to WHO recommendations.

Some question from the Methodology chapter:

1. What criteria were used to match the cases and control groups? You wrote, "matched by propensity score matching". Please describe the factors that have been involved, and indicate the sources of literature where this sampling method would be used or described. Is age, gender, education, etc. taken into account when matching the groups?

2. Please explain in more detail what kind of questionnaires (24h-recall, FFQ, etc.) have been used to assess dietary habits.

A high-fat diet needs to be described in more detail. How was this diet calculated? How much energy per day from fats received responders? What kind of criteria were used to identify the high-fat diet of respondents? It is also necessary to refer to the sources of literature.

It is extremely important to describe this because you have obtained meaningful results and presented them in the conclusions.

3. It is necessary to provide literature references or recommendations that you have used to evaluate overweight, obesity, and the criteria for WC and dyslipidemias.

Results section

In Table 1 you presented, that only total cholesterol values significantly differed in cases compared to controls, however, neither did TG, LDL-c, or HDL-C, but the incidence of dyslipidemias was significantly different. How do you explain this?

Are the data in this table adjusted for gender and age?

Please provide the full p values in Table 2. In my opinion, some presented data are not significantly different.

Discussion section.

More attention should be paid to the interaction between risk factors for CHD and to comparisons these results with researches in other countries. On the other hand, it is difficult for the reader to compare those results because of the different evaluation criteria and the uncertainty of what a high-fat diet represents (EU is considered to be a high-fat diet if the daily energy from fat is more than 65% per day).

The conclusion is suitable and reflects on the aim.

Literature should be updated with more recent sources, as only 40% of sources are under 5 years old.

The English version of the article should be improved.

Reviewer #2: The study work sounds fine, the number of included subjects and the study design give a strength to your study. i have a few comments:

- I noticed many repetitive sentences/words in the draft, maybe due to mistyping, need to be removed( e.g., last phrase in the statistical analysis paragraph).

2- i didn't any information about the malignancy history ( cancer, chemo/radio HX) of those patients or if any of them has any active malignancy at the time of enrollment, and if so, you didn't clarify if those are included in your study or not.

reason for mentioning that is, currently attention is paid more to cardio-oncology and enormous of studies have proven that active malignancy/chemo/radiotherapy or a history of any, increase the risk of CVD, in that case you will need to consider that in you analysis as a possible confounders. Same thing for their psychiatric history.

Thanks

6. PLOS authors have the option to publish the peer review history of their article (what does this mean?). If published, this will include your full peer review and any attached files.

Reviewer #1: No

Reviewer #2: No

---

## [Author Response · Author response to Decision Letter 0]

13 Jan 2020

Dear Alana T Brennan and Reviewers:

Many thanks for your interest in our manuscript entitled “Risk of coronary heart disease among Xinjiang rural: a nested case-control study in China” (ID: PONE-D-19-21029) for publication. We appreciate the thoughtful comments from the reviewers, and we have made revisions accordingly. We outline in the following text a detailed point-by-point response to the comments and the corresponding manuscript revisions. After careful consideration, we decided used the multiplicative interaction index as an effective indicator to evaluate the multiplicative interaction, which was revised in the manuscript. We have uploaded a database used for data analysis and our questionnaire as the Supporting Information.

We hope that the revised version of the manuscript is now acceptable for publication in your journal. I look forward to hearing from you soon.

With best wishes,

Yours sincerely,

Shuxia Guo, PhD

Corresponding author

Department of Public Health, Shihezi University School of Medicine, Shihezi, Xinjiang 832000, China

Phone: +86-180-0993-2625; Fax: +86-0993-2057-153; Email: gsxshzu@sina.com

We would like to express our sincere thanks to the reviewers for the constructive and positive comments.

Reviewer #1:

Some question from the Methodology chapter:

1. Response to comment: What criteria were used to match the cases and control groups? You wrote, "matched by propensity score matching". Please describe the factors that have been involved, and indicate the sources of literature where this sampling method would be used or described. Is age, gender, education, etc. taken into account when matching the groups?

Response: We are very sorry for not descript this part clearly, we have rewritten this part as requested by the reviewers in Methods (Selection of cases and controls). 

Methods: For each case, we selected two controls, which were matched by gender, age, and ethnicity used propensity score [1].

1. Lin HC, Xirasagar S, Lee C-Z, Huang C-C, Chen C-H: The association between gastro-oesophageal reflux disease and subsequent rheumatoid arthritis occurrence: a nested case–control study from Taiwan. Bmj Open, 7(11): e016667.

2.1 Response to comment: Please explain in more detail what kind of questionnaires (24h-recall, FFQ, etc.) have been used to assess dietary habits.

Response: We used the self-made questionnaire of national natural fund project (No.81560551) to assess dietary habits by food frequency and the definition of a high-fat diet is also written into the Method. Here are the survey questionnaire options for high-fat foods:

82. I18 Zhuafan (1 serving is equivalent to 1 bowl (8cm caliber) or 1 plate (16cm caliber)): 

○ Do not eat or less than 1 serving per week

○ 1-3 servings per week

○ 4-6 servings per week

○ 1 serving per day

○ 2 servings or more per day

83. I19 Fried Food (Buersake):

○ Do not eat or less than once a week

○ 1-3 times a week

○ 4-6 times a week

○ Once a day

○ 2 times or more per day

84. I20 Ghee: 

○ No

○ 1-2 scoops per day

○ 3-4 scoops per day

○ 5-6 scoops per day

○ 6 scoops and above

Definition: Local high–fat foods include the following: Zhuafan is stewed with fresh lamb, carrots, rice, onions, mutton fat and a large amount of oil (approximately 371 kcal per 100 gm); Baoershake is a fried flour–based food item (approximately 399 kcal per 100 g,); and ghee, a local dairy product similar to butter, is extracted from milk and contains more than 90% saturated fat (approximately 860 kcal per 100 gm), and is always consumed with naan and milk tea. We defined frequent high-fat food consumption as conformance with any of the following criteria: (1) eating 4 servings or more of Zhuafan per week (1 serving is equivalent to 1 bowl of 8–cm caliber or 1 plate of 16–cm caliber), (2) eating Baoershake 4 times or more per week, or (3) eating 5 scoops or more of ghee per day.

2.2 Response to comment: A high-fat diet needs to be described in more detail. How was this diet calculated? How much energy per day from fats received responders? What kind of criteria were used to identify the high-fat diet of respondents? It is also necessary to refer to the sources of literature.

It is extremely important to describe this because you have obtained meaningful results and presented them in the conclusions.

Response: Regarding the definition of a high-fat diet, we asked individuals about their eating habits (including ghee, Zhuafan, Buersake) through interview questionnaires, but we can define them only by the frequency of eating the food. We also describe it in the Definition of Methods. To make it easier for readers to understand the meaning of the indicators we used, we changed the high-fat diet to frequent high-fat food consumption.

3. Response to comment: It is necessary to provide literature references or recommendations that you have used to evaluate overweight, obesity, and the criteria for WC and dyslipidemias.

Response: We have added literature references to evaluate overweight, obesity, and the criteria for WC and dyslipidemias.

4. Response to comment: 

Results section

（1）In Table 1 you presented, that only total cholesterol values significantly differed in cases compared to controls, however, neither did TG, LDL-c, or HDL-C, but the incidence of dyslipidemias was significantly different. How do you explain this?

（2）Are the data in this table adjusted for gender and age?

Response: (1) After we analyzed and obtained the following results, which might due to the sample size of TG, LDL-c or HDL-C are not sufficient to show the statistical differences. When the four types are integrated into one indicator (dyslipidemia), the sample size is sufficient to show the significant statistical differences (P < 0.05). In addition, it may be caused by different sensitive cut-off points of TG, LDL-c or HDL-C between different ethnic groups and further analysis is requested in later studies.

 (2) We apologize for the confusion caused by the lack of a clear description. Since we have considered age, gender, and ethnicity in the propensity score matching, we used this as a rule to match two control groups without CHD for each case. We have noted the confounding factors adjusted by each model in the footer of the table.

5. Response to comment: Please provide the full p values in Table 2. In my opinion, some presented data are not significantly different.

Response: We are sorry that Andersson’s Excel calculation table did not provided the P values of the three indicators. This calculation table can only provide the index values and their 95% confidence intervals. In addition, the 95% confidence interval can be used as an effective indicator to distinguish whether statistical differences exist or not. 

6. Response to comment: 

Discussion section.

More attention should be paid to the interaction between risk factors for CHD and to comparisons these results with researches in other countries. On the other hand, it is difficult for the reader to compare those results because of the different evaluation criteria and the uncertainty of what a high-fat diet represents (EU is considered to be a high-fat diet if the daily energy from fat is more than 65% per day).

Response: We have compared the interaction of coronary heart disease factors with the results of domestic and foreign studies in Discussion. However, due to the lack of reports on the interaction of influencing factors involved in this article, we have selected as many representative literatures as possible for comparison and explanation. We also mentioned this in the discussion.

Our research on frequent high-fat food consumption is intended to provide some clues, which is necessary to be discussed but limited when compared with the domestic and international literature, so we write that in the Limitation.

Limitation: Third, we did not measure the ratio of fat in the daily total energy and daily dietary fat intake to evaluate the high–fat food consumption; we only used the frequency of high–fat food consumption as clues.

7. Response to comment: The conclusion is suitable and reflects on the aim.

Literature should be updated with more recent sources, as only 40% of sources are under 5 years old.

Response: We have updated our sources on the basis of some earlier literature that must be cited, 60% of references are within five years.

8. Response to comment: The English version of the article should be improved.

Response: We have contacted the editing company to further improve the language of our article.

Reviewer #2: 

1. Response to comment: I noticed many repetitive sentences/words in the draft, maybe due to mistyping, need to be removed (e.g., last phrase in the statistical analysis paragraph).

Response: We checked the manuscript and rewritten some sentences based on the reviewer's suggestions.

2. Response to comment: I didn't any information about the malignancy history (cancer, chemo/radio HX) of those patients or if any of them has any active malignancy at the time of enrollment, and if so, you didn't clarify if those are included in your study or not. 

reason for mentioning that is, currently attention is paid more to cardio-oncology and enormous of studies have proven that active malignancy/chemo/radiotherapy or a history of any, increase the risk of CVD, in that case you will need to consider that in your analysis as a possible confounder. Same thing for their psychiatric history.

Thanks

Response: Thank you very much for the suggestions of the reviewers, which made our study more complete. After the verification of social security data, we found that there were no patients with psychiatric in this nested case-control study and only 5 patients in the control group had history of cancer. Because the number of malignant tumors is small, it cannot be adjusted as a confounding factor. Of course, this should be used as an exclusion criterion, although we did not consider it in the previous design. Thus, we put that in the Limitation.

Limitation and strength: Fourth, history of malignancy was not excluded. Despite these limitations, the study also has some strengths.

---

## [Editor Report · Decision Letter 1]

22 Jan 2020

PONE-D-19-21029R1

Risk of coronary heart disease in the rural population in Xinjiang: a nested case-control study in China

PLOS ONE

Dear Dr. Guo,

Thank you for submitting your manuscript to PLOS ONE. After careful consideration, we feel that it has merit but does not fully meet PLOS ONE’s publication criteria as it currently stands. Therefore, we invite you to submit a revised version of the manuscript that addresses the points raised during the review process.

After further review of your manuscript there are a couple points that need clarification.

1. You state that you matched on gender and age and used propensity scores. It is unclear to me if you matched on gender, age and other variables in your propensity score OR if gender and age were the only variables that were in your propensity score. If it is the latter then I am unclear as to why you used a propensity score to do the matching. Also, it is good practice to show if the matching actually worked by showing the distribution of the matched factors between cases and controls, which you do not do. It is also good practice to show a histogram of the propensity scores stratified by cases and controls. These can be added as supplemental figures to your paper. Please address these issues.

2. You assess multiplicative interaction in your study. It is well known that most studies are under powered to assess interaction appropriately. You do not discuss this as a limitation in your paper. Please address this.

We would appreciate receiving your revised manuscript by February 15, 2020. To enhance the reproducibility of your results, we recommend that if applicable you deposit your laboratory protocols in protocols.io, where a protocol can be assigned its own identifier (DOI) such that it can be cited independently in the future. For instructions see: http://journals.plos.org/plosone/s/submission-guidelines#loc-laboratory-protocols

We look forward to receiving your revised manuscript.

Kind regards,

Alana T Brennan

Academic Editor

PLOS ONE

Journal Requirements:

Additional Editor Comments (if provided):

1. You state that you matched on gender and age and used propensity scores. It is unclear to me if you matched on gender, age and other variables in your propensity score OR if gender and age were the only variables that were in your propensity score. If it is the latter then I am unclear as to why you used a propensity score to do the matching. Also, it is good practice to show if the matching actually worked by showing the distribution of the matched factors between cases and controls, which you do not do. It is also good practice to show a histogram of the propensity scores stratified by cases and controls. These can be added as supplemental figures to your paper. Please address these issues.

2. You assess multiplicative interaction in your study. It is well known that most studies are under powered to assess interaction appropriately. You do not discuss this as a limitation in your paper. Please address this.

---

## [Author Response · Author response to Decision Letter 1]

28 Jan 2020

Dear Alana T Brennan:

Many thanks for your interest in our manuscript entitled “Risk of coronary heart disease among Xinjiang rural: a nested case-control study in China” (ID: PONE-D-19-21029) for publication. We would like to express our sincere thanks to the reviewers and editor for the constructive comments, and we have made revisions accordingly. We outline in the following text a detailed point-by-point response to the comments and the corresponding manuscript revisions. 

We hope that the revised version of the manuscript is now acceptable for publication in your journal. I look forward to hearing from you soon.

With best wishes,

Yours sincerely,

Shuxia Guo, PhD

Corresponding author

Department of Public Health, Shihezi University School of Medicine, Shihezi, Xinjiang 832000, China

Phone: +86-180-0993-2625; Fax: +86-0993-2057-153; Email: gsxshzu@sina.com

Journal Requirements:

Additional Editor Comments (if provided):

1. You state that you matched on gender and age and used propensity scores. It is unclear to me if you matched on gender, age and other variables in your propensity score OR if gender and age were the only variables that were in your propensity score. If it is the latter then I am unclear as to why you used a propensity score to do the matching. Also, it is good practice to show if the matching actually worked by showing the distribution of the matched factors between cases and controls, which you do not do. It is also good practice to show a histogram of the propensity scores stratified by cases and controls. These can be added as supplemental figures to your paper. Please address these issues.

Response:

In this paper, nested case-control study design was adopted. By consulting the nested case-control literature, we found that many authors used propensity score to match case groups to control groups, and we also cited the articles. In addition, propensity score is a good matching method, so we chose propensity score to match case groups to control groups. We uploaded the file of “Description of matching factors between Case and Control by SPSS” to show the distribution of gender, age and ethnicity between the case group and the control group in the Supporting file.

2. You assess multiplicative interaction in your study. It is well known that most studies are under powered to assess interaction appropriately. You do not discuss this as a limitation in your paper. Please address this.

Response: After discussion, we decided to remove this part from the limitations and advantages.

---

## [Editor Report · Decision Letter 2]

3 Feb 2020

PONE-D-19-21029R2

Risk of coronary heart disease in the rural population in Xinjiang: a nested case-control study in China

PLOS ONE

Dear Dr. Guo,

Thank you for submitting your manuscript to PLOS ONE. After careful consideration, we feel that it has merit but does not fully meet PLOS ONE’s publication criteria as it currently stands. Therefore, we invite you to submit a revised version of the manuscript that addresses the points raised during the review process.

1. The figures that you create for the propensity score should be add in as supplemental tables and figures. Please edit them accordingly so that they are journal ready.

2. One last clarification is need in regards to the patient population. You stated in the study design that, "The survey among the Kazakhs and the Uyghurs began in April 2009 and was followed up in December 63 2013, April 2016, and August 2017 in the Xinyuan County and Jiashi County."  and then you state that, "The total response rate was 87.5% (88.6% among the 70 Uygur and 86.5% among the Kazakh)." Can you clarify that the patient population was the same each year - 2013, 2016 and 2017? I also assume that the rates of loss differed each year? You need to discuss the roughly 13% loss to follow-up as a limitation in your discussion. Is there reason to believe these ~13% of patients differ to the ones that were included? Please discuss this in your limitations section as it suggests potential selection bias.

3. In your response to my question in the last review, "2. You assess multiplicative interaction in your study. It is well known that most studies are under powered to assess interaction appropriately. You do not discuss this as a limitation in your paper. Please address this." you stated. "Response: After discussion, we decided to remove this part from the limitations and advantages." You should have addressed this as a limitation not remove it from your discussion. Please fix this.

We would appreciate receiving your revised manuscript by 2/15/2020. To enhance the reproducibility of your results, we recommend that if applicable you deposit your laboratory protocols in protocols.io, where a protocol can be assigned its own identifier (DOI) such that it can be cited independently in the future. For instructions see: http://journals.plos.org/plosone/s/submission-guidelines#loc-laboratory-protocols

We look forward to receiving your revised manuscript.

Kind regards,

Alana T Brennan

Academic Editor

PLOS ONE

Additional Editor Comments (if provided):

1. The figures that you create for the propensity score should be add in as supplemental tables and figures. Please edit them accordingly so that they are journal ready.

2. One last clarification is need in regards to the patients consented. You stated in the study design that, "The survey among the Kazakhs and the Uyghurs began in April 2009 and was followed up in December 63 2013, April 2016, and August 2017 in the Xinyuan County and Jiashi County." and then you state that, "The total response rate was 87.5% (88.6% among the 70 Uygur and 86.5% among the Kazakh)." Can you clarify that the patient population was the same each year - 2013, 2016 and 2017? I also assume that the rates of loss differed each year? You need to discuss the roughly 13% loss to follow-up as a limitation in your discussion. Is there reason to believe these ~13% of patients differ to the ones that were included? If so then you need to discuss this in your limitations section as it suggests potential selection bias.

3. In your response to my question in the last review, "2. You assess multiplicative interaction in your study. It is well known that most studies are under powered to assess interaction appropriately. You do not discuss this as a limitation in your paper. Please address this." you stated. "Response: After discussion, we decided to remove this part from the limitations and advantages." You should have addressed this as a limitation not remove it from your discussion. Please fix this.

---

## [Author Response · Author response to Decision Letter 2]

9 Feb 2020

Response Letter

Dear Alana T Brennan:

Many thanks for your interest in our manuscript entitled “Risk of coronary heart disease among Xinjiang rural: a nested case-control study in China” (ID: PONE-D-19-21029) for publication. We appreciate the points raised during the review process, and we have made revisions accordingly. We outline in the following text a detailed point-by-point response to the comments and the corresponding manuscript revisions. 

We hope that the revised version of the manuscript is now acceptable for publication in your journal. I look forward to hearing from you soon.

With best wishes,

Yours sincerely,

Shuxia Guo, PhD

Corresponding author

Department of Public Health, Shihezi University School of Medicine, Shihezi, Xinjiang 832000, China

Phone: +86-180-0993-2625; Fax: +86-0993-2057-153; Email: gsxshzu@sina.com

Journal Requirements:

Additional Editor Comments (if provided):

1. The figures that you create for the propensity score should be add in as supplemental tables and figures. Please edit them accordingly so that they are journal ready.

Response: We have edited the figures and add in Supporting Information as supplemental tables and figures.

2. One last clarification is need in regards to the patients consented. You stated in the study design that, "The survey among the Kazakhs and the Uyghurs began in April 2009 and was followed up in December 63 2013, April 2016, and August 2017 in the Xinyuan County and Jiashi County." and then you state that, "The total response rate was 87.5% (88.6% among the 70 Uygur and 86.5% among the Kazakh)." Can you clarify that the patient population was the same each year - 2013, 2016 and 2017? I also assume that the rates of loss differed each year? You need to discuss the roughly 13% loss to follow-up as a limitation in your discussion. Is there reason to believe these ~13% of patients differ to the ones that were included? If so then you need to discuss this in your limitations section as it suggests potential selection bias

Response: Thank you very much for your constructive suggestions. In this manuscript, we describe an overall response rate of 87.5%, which means that the baseline population have been followed up accounted for 87.5% of the total baseline population after three follow-ups, and the response rates were different at each follow-up (2013, 2016 and 2017). And 12.5% of the baseline population have not been followed up and their disease status cannot be confirmed, which may cause potential selection bias. Therefore, we added this section to the Limitations.

Limitations: And the overall response rate in our study was 87.5%, which means that the baseline population have not been entirely followed up, and we cannot confirm the disease status of the baseline population who were not followed up. This means that there may be selection bias.

3. In your response to my question in the last review, "2. You assess multiplicative interaction in your study. It is well known that most studies are under powered to assess interaction appropriately. You do not discuss this as a limitation in your paper. Please address this." you stated. "Response: After discussion, we decided to remove this part from the limitations and advantages." You should have addressed this as a limitation not remove it from your discussion. Please fix this.

Response: We are very sorry that we misunderstood your question. We have added this part in the Limitations.

Limitations: we did not appropriately evaluate all the effects of interactions between factors on CHD.

---

## [Editor Report · Decision Letter 3]

11 Feb 2020

Risk of coronary heart disease in the rural population in Xinjiang: a nested case-control study in China

PONE-D-19-21029R3

Dear Dr. Guo,

We are pleased to inform you that your manuscript has been judged scientifically suitable for publication and will be formally accepted for publication once it complies with all outstanding technical requirements.

With kind regards,

Alana T Brennan

Academic Editor

PLOS ONE
---

## [Editor Report · Acceptance letter]

19 Feb 2020

PONE-D-19-21029R3 

Risk of coronary heart disease in the rural population in Xinjiang: a nested case-control study in China 

Dear Dr. Guo:

I am pleased to inform you that your manuscript has been deemed suitable for publication in PLOS ONE. Congratulations! Your manuscript is now with our production department. 

With kind regards,

on behalf of

Dr. Alana T Brennan 

Academic Editor

PLOS ONE